# UniCoD: Enhancing Robot Policy via Unified Continuous and Discrete Representation Learning

## Abstract

Building generalist robot policies that can handle diverse tasks in open-ended environments is a central challenge in robotics. To leverage knowledge from large-scale pretraining, prior work has typically built generalist policies either on top of vision-language understanding models (VLMs) or generative models. However, both semantic understanding from vision-language pretraining and visual dynamics modeling from visual-generation pretraining are crucial for embodied robots. Recent unified models of generation and understanding have demonstrated strong capabilities in both comprehension and generation through large-scale pretraining. We posit that robotic policy learning can likewise benefit from the combined strengths of understanding, planning and continuous future representation learning. Building on this insight, we introduce UniCoD, which acquires the ability to dynamically model high-dimensional visual features through pretraining on over 1M internet-scale instructional manipulation videos. Subsequently, UniCoD is fine-tuned on data collected from the robot embodiment, enabling the learning of mappings from predictive representations to action tokens. Extensive experiments show our approach consistently outperforms baseline methods in terms of 9% and 12% across simulation environments and real-world out-of-distribution tasks. Demos and code can be found at our anonymous website.

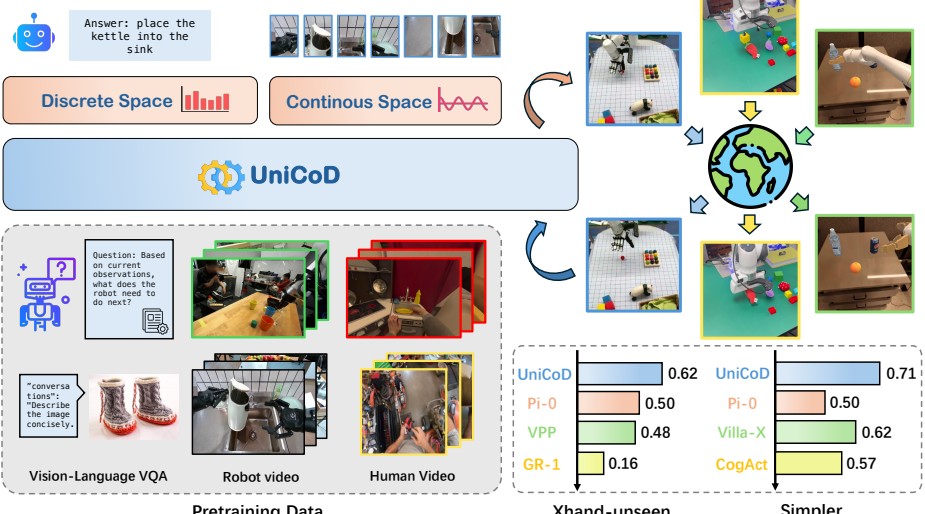

Figure 1: **Overview of UniCoD.** Our proposed UniCoD, which utilizes both understanding and prediction tasks under discrete and continuous representation space, demonstrates strong semantic generalization capabilities on real-world manipulation tasks, particularly in its ability to handle completely novel objects not seen during training. The upper right displays benchmark evaluations across several simulations and 2 real-world robots.

# 1 INTRODUCTION

Constructing generalist foundation models (Zitkovich et al., 2023; Kim et al., 2024b) for robots manipulation in the physical world has emerged as a rapidly growing frontier within embodied AI. Vision–language–action (VLA) models aim to learn robotic policies from data annotated with vision, linguistic, and action signals. However, the scarcity of robotic data and the heterogeneity across embodiments present substantial challenges, particularly in achieving generalization to novel scenes and task instructions, and in accurately predicting actions.

To mitigate these limitations, recent studies have explored mapping Vision–Language Models (VLMs) into the action space(Black et al., 2024; Team et al., 2024). This strategy provides robot policies with alignment priors across language and vision modalities. Nevertheless, these approaches often overlook the fundamental discrepancies between robotic action tasks and vision–language tasks. Unlike the abundance of internet-scale vision–language data, fine-tuning VLMs on limited robotic datasets frequently leads to degradation of their foundational capabilities (Xing et al., 2025). Complementary lines of work have investigated leveraging generation models as intermediaries for action policy learning(Hu et al., 2024; Wen et al., 2024). While such visual foresight approaches facilitate dynamic representation learning and enable the use of heterogeneous data sources, they typically fail to preserve vision–language alignment inherent to pretrained VLMs. These observations highlight a central insight: it is crucial to design robot-specific post-training paradigms tailored to embodied scenarios. Upon re-examining this line of approaches, we observe that both language understanding and future state prediction can provide preliminary guidance for general manipulation tasks. The unified learning strategy further enables the model to acquire representations beneficial for robotic tasks from a broader range of data.

Building upon these insights and prior advances in vision–language–action (VLA) research(Zhang et al., 2025; Wang et al., 2025b), we propose UniCoD, which follows an understanding–generation–execution paradigm that integrates discrete task comprehension with continuous prediction of future robotic states. To address heterogeneous modalities, UniCoD employs a MOT architecture (Liang et al., 2024) with modality-specialized experts. UniCoD is trained in two stages to introduce continuous feature forecasting to action learning while maintaining general capabilities within VLM. In the first stage, we curate and label a diverse collection of Embodied QA data sourced from both robots(Khazatsky et al., 2024; Bu et al., 2025; Wu et al., 2024) and human demonstrations(Hoque et al., 2025; Grauman et al., 2022). We enable the model to learn discrete language representations for understanding of embodied scenes and continuous visual representations for world modeling. In the second stage, we introduce embodiment-specific robotic data annotated with action behaviors. By jointly predicting continuous visual futures and actions, the model learns to utilize semantically aligned features that are rich in dynamic information. This, in turn, equips the `VLA` policy with better generalization capabilities for new objects and scenes.

In experiments, UniCoD achieves a 9% improvement in the Simpler benchmark compared to existing SOTA approach and demonstrates strong semantic generalization for real-world robots for complex tasks on both robot arms and dexterous hands. In summary, our contributions are as follows:

• We propose a novel vision–language–action (VLA) that integrates both discrete and continuous representations for understanding and learning dynamics, which is pre-trained on large-scale data from both robot and human demonstrations, enabling effective transfer to embodied tasks.

• We propose a two-stage training framework that aligns action representations while preserving the aligned intermediate representations.

• Our best-performing model achieves state-of-the-art results across both simulated and real-world environments, and we further analyze the impact of different feature design choices on the model's capabilities.

# 2 RELATED WORKS

**Vision-Language-Action Models**  Vision-Language-Action (VLA) models introduce multimodal large language models (Dai et al., 2024; Touvron et al., 2023; Wang et al., 2025a; Bai et al., 2025) into robot policy models to enhance their generalization ability (Brohan et al., 2023; Kim et al., 2024a; Black et al., 2024; Guo et al., 2025). This line of work either utilizes the VLM and an

action head for end-to-end action prediction (Li et al., 2023; Wen et al., 2025) or uses the VLM to extract key information to condition downstream policy (Zhang et al., 2024; Li et al., 2025). Some recent works have introduced additional auxiliary tasks to VLAs, including enhancing spatial understanding (Qu et al., 2025), QA reasoning (Zhou et al., 2025), visual reasoning (Zhao et al., 2025) and prediction (Zhang et al., 2025), demonstrating that both general-purpose understanding and generation capabilities can promote action learning. However, these methods are primarily limited to unifying generative tasks within a discrete token prediction framework, which may compromise the robust vision-language alignment inherent in the pre-trained VLM. In this work, we incorporate a continuous-space visual prediction task to aid downstream action learning.

**Generalist Robot Policies with Joint Prediction**   Explorations into generalist robot policies have considered using world models (Blattmann et al., 2023; Assran et al., 2025; Chen et al., 2024; Guo et al., 2024) to learn physical dynamics and subsequently predict actions (Du et al., 2024; Black et al., 2023). Many recent methods have incorporated prediction into larger-scale data and models: GR-1 (Wu et al., 2023) utilizes video pre-training to initialize the action policy; VPP (Hu et al., 2024) uses a video foundation model as the visual encoder for action policy. While these methods fully leverage the rich information from video data, they lack semantic grounding capabilities due to the absence of large language models. Recent works (Zhang et al., 2025; Wang et al., 2025b) use VQ quantization to incorporate predictive generation tasks into VLA policies, demonstrating the potential for unifying understanding and prediction. In contrast, we utilize continuous visual features as the prediction supervision signal and pre-train our model on large-scale language prediction and continuous visual prediction tasks.

# 3   METHODOLOGY

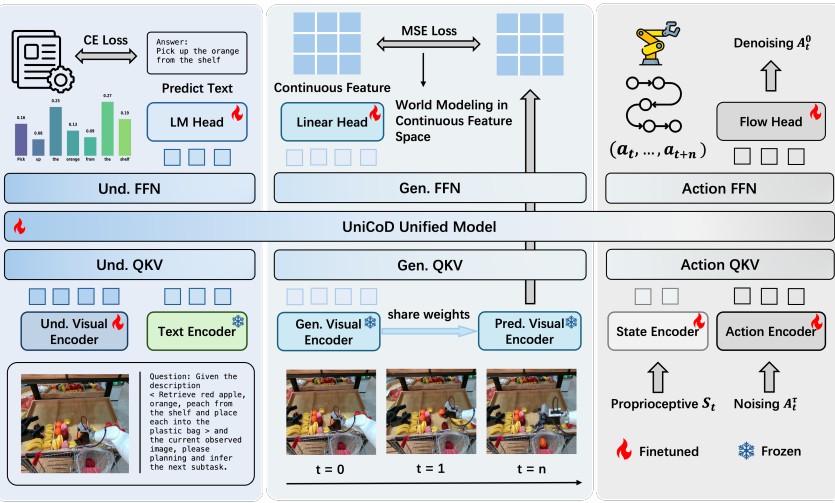

Figure 2: **Illustration of the UniCoD framework.** UniCoD adopts a MoT framework to handle text understanding and planning, continuous visual prediction, and action execution. The continuous features are derived from future observations using a frozen vision encoder.

In this section, we present the overall framework design and the two-stage training strategy of UniCoD, as illustrated in Figure 2. In the first stage, UniCoD is trained to learn joint text–image representations across diverse manipulation datasets, including understanding, planning, and continuous future prediction tasks. In the subsequent stage, an action expert is employed to integrate the multimodal inputs and predicted future states with action. In the subsequent subsections, we will respectively describe: (1) the joint visual-language embedding learning for pre-training in Sec 3.1, (2) our policy learning method in Sec 3.2, and (3) the implementation details and training data in Sec 3.3.

## 3.1 Unified vision language joint embedding modeling

Before introducing the robot action space, we first establish a cross-embodiment pre-training paradigm for robots. In this stage, a subset of the model parameters $U_{v,l}$ is jointly optimized via the Text-Image to Embedding(TI2E) (examples can be found in A.5). Concretely, given a language instruction $l$ and the current view observations $o_t$ at time $t$, UniCoD is trained to predict the joint visual–text embedding: $\hat{o}_{t+h}, \hat{l} = U_{v,l}(o_t, l)$, where $\hat{o}_{t+h} = V(o_{t+h}) = \{c_1, c_2, \ldots, c_n\}$ denotes the predicted continuous future representation encoded by the visual encoder $V$, while $\hat{l} = \{d_1, d_2, \ldots, d_m\}$ corresponds to the $m$-token textual sequence.

**Discrete Representation Learning.** To enhance vision–language alignment, the parameters $U_{v,l}$ are initialized from a pre-trained vision–language model. Fine-grained language representations are derived from large-scale vision–language datasets, as well as planning and scene descriptions from embodied tasks, which are annotated using pre-trained MLLM into a VQA-style format. This target enables the agent to gain a better understanding of diverse instructions and scenes, thereby facilitating the learning of continuous representations for visual prediction and action.

**World Modeling under Continuous Space.** In the pre-training stage, to acquire dynamic representations associated with the action space, we introduce additional attention weights dedicated to future state prediction, which are integrated with the original VLM within the mixture-of-transformers framework. Unlike prior approaches that directly predict image pixels, we leverage a frozen visual encoder to represent future observations in a continuous high-dimensional space, capturing high-level information across different semantics. A more detailed discussion can be found in Appendix A.1.

For the visual inputs, we employ a dual-encoder design that combines the VLM visual encoder with a generator encoder. The tokens generated by the latter are processed by the generative expert in the mixture-of-transformers and, together with the language tokens and VLM visual tokens, jointly participate in the attention computation. This design preserves the pretrained model's vision–language alignment while enabling the prediction process to benefit from richer semantic understanding.

**Training Objective.** The visual and language inputs are processed respectively through the MoT framework, then autoregresively generate $\hat{l}_{t+h}^{pred} = d_{1:m}^{pred}$, while the generation expert obtains the $\hat{o}_{t+h}^{pred} = c_{1:n}^{pred}$. We follow the standard setup of generative–understanding models, employing cross-entropy loss for the language branch and mean squared error loss for the generative branch. This optimaze progress can be formulated as:

$$\mathcal{L}_1 = \lambda_1 \cdot \frac{1}{n} \sum_{i=1}^{n} \left\| c_i^{pred} - c_i \right\|_2^2 - (1 - \lambda_1) \cdot \frac{1}{m} \sum_{j=1}^{m} log P_\theta(d_j \mid d_{<j}, l, o_t) \tag{1}$$

where $\lambda_1$ serves as a weighting factor to balance the loss contributions of the discrete and continuous representations.

## 3.2 Unified action modeling

In the previous stage, we obtained $U_{v,l}$ through pre-training, which endowed the model with basic capabilities in future state prediction and vision–language alignment. However, $U_{v,l}$ cannot yet be directly mapped to the action space. To address this limitation, in the second stage we fine-tune $U_{v,l}$ on embodiment data comprising visual, language, and action modalities, while simultaneously training an action expert from scratch to construct $U_{v,l,a}$.

**Action & State Expert.** Similar to the generation and understanding experts, we employ distinct attention weights to project actions and states (i.e., proprioception) into a shared attention space. Unlike the other experts, the action expert leverages flow matching to capture the continuous and inherently multi-modal distribution of the action space. Proprioceptive signals $s_t$ are processed by an MLP-based state expert encoder, enabling fusion within the unified model. Given an action sequence $A_t = (a_t, a_{t+1}, \ldots, a_{t+h})$ to be executed, along with the observation $o_t$ and instruction $l$, the unified model $U_{v,l,a}$ is trained to approximate vector fields as:

$$\mathcal{L}_{\text{flow}} = \mathbb{E}_{\tau \sim \mathcal{U}(0,1)} \mathbb{E}_{\{A_t, o_t, s_t, l\} \sim \mathcal{D}} \left[ \| U_{v,l,a}(A_t^\tau, o_t, s_t, l, \tau) - (A_t - A_t^\tau) \|_2^2 \right], \tag{2}$$

where $A_t^\tau = (1 - \tau)\epsilon + \tau A_t$ denotes the interpolated actions at step $\tau$, and $\epsilon \sim \mathcal{N}(0, I)$.

In this action training stage, we also jointly optimize the generation expert by predicting the future observation states $c_{1:n}$, yielding the following objective:

$$\mathcal{L}_2 = \lambda_2 \cdot \frac{1}{n} \sum_{i=1}^{n} \left\| c_i^{\text{pred}} - c_i \right\|_2^2 + (1 - \lambda_2)\mathcal{L}_{\text{flow}}. \tag{3}$$

## 3.3 IMPLEMENTATION DETAILS

**Model Setting.** UniCoD employs Paligemma Beyer et al. (2024) as the VLM expert. For future observation encoding, we experiment with SigLIP Tschannen et al. (2025), DINOv3 Siméoni et al. (2025), and direct pixel-level prediction. Considering the information flow across modalities, we adopt a block-wise masking mechanism in the MoT attention: within each modality, bidirectional attention is applied, while across modalities a causal mask is enforced following the order of image, language, image prediction, state information, and action.

**Pre-training Data.** In the pretraining stage, we utilize three categories of data to acquire joint text–image representations: (1) 320k robot videos paired with fine-grained subtask descriptions and overall task instructions, which yield VQA and TI2E data for the generation–understanding task; (2) 870k robot and human operation videos accompanied by task instructions, which are used as TI2E data; and (3) 560k generic vision–language question answering data, employed for co-training to preserve the fundamental capabilities of the VLM. In the action modeling stage, we exclusively adopt VLA data collected in both simulation and real-world robotic environments. Further details regarding the datasets are provided in Appendix A.5.

## 4 EXPERIMENT

To comprehensively evaluate our proposed method, UniCoD, we conduct extensive experiments across two simulation benchmarks and on two distinct real-world robotic platforms. Our experiments are designed to assess the performance of UniCoD and validate the effectiveness of our proposed modules.

## 4.1 EXPERIMENTAL SETUP

Our experiments are conducted and deployed across four distinct environments. Figure 3 illustrates a selection of tasks from both our simulation and real-world settings.

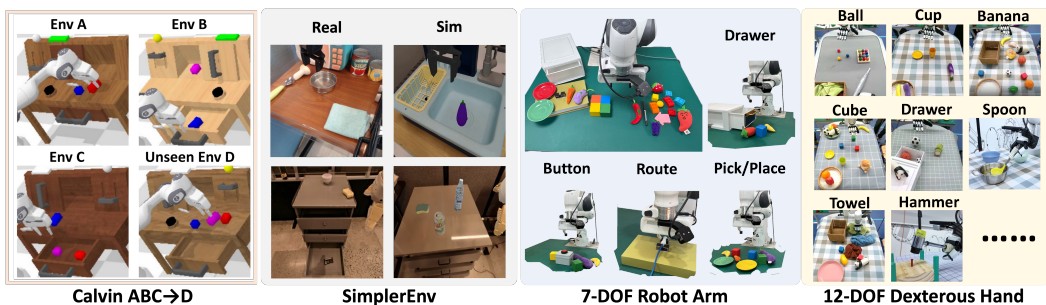

Figure 3: Our evaluation environments, including 2 simulation benchmarks and 2 real-world embodiments.

**Calvin Benchmark** Calvin is a simulation benchmark designed for evaluating long-horizon, language-conditioned manipulation policies. We employ the *ABC-D* split to evaluate the single-view generalization capabilities of the models. The evaluation suite includes 1,000 long-horizon sequences, each of length 5. We report the average length of completed sub-task sequences.

**SimplerEnv Benchmark** SimplerEnv is a simulation benchmark designed to evaluate policies trained on real-world datasets, such as Bridge-V2 and Fractal. The benchmark supports two types of robot arms: WindowX and Google Robot. For our evaluation, we conduct 240 runs for each task and report the average success rate.

**Real-World Franka Emika Panda Arm**   We deploy models on a Franka Emika arm for real-world task comparison. We first collected a dataset of 2,000 trajectories spanning over 20 distinct tasks, encompassing six fundamental skills: picking, placing, opening a drawer, closing a drawer, pressing a button, and routing a cable. We evaluate performance on both seen and unseen task variations. The unseen category primarily involves grasping novel objects not present in the training data and introducing misleading objects. More details can be found in Appendix A.3.1.

**Real-World XArm with 12-DOF X-Hand**   On our dexterous manipulation platform, we train different models using a dataset of 4,000 trajectories across more than 100 tasks. The models are then evaluated in a variety of seen and unseen scenarios, which cover 13 distinct skills in 9 categories. More details can be found in Appendix A.3.2.

## 4.2   SIMULATION EXPERIMENTS

**Implementation Details**   We first pre-train UniCoD following the methodology described in Section 3. Subsequently, we fine-tune the model on 8 A100 GPUs for 22k steps, using a learning rate of $5 \times 10^{-5}$ and a batch size of 1024. For all simulation training, we consistently use a single, third-person-view image of size $224 \times 224$ as the visual input. In Calvin, we use an action chunk size of 10, and during deployment, the full 10-step chunk is executed at each inference step. In SimplerEnv, we use an action chunk size of 4; for the WindowX environment (corresponding to the Bridge dataset), the full 4-step chunk is executed, whereas for the Google Robot environment (corresponding to the Fractal dataset), half of the action chunk is executed.

**Baselines**   We compare UniCoD against several state-of-the-art VLAs and prediction-based policies. On SimplerEnv, we benchmark UniCoD against RT-1-X (Brohan et al., 2022), Octo (Team et al., 2024), OpenVLA (Kim et al., 2024a), RoboVLMs (Liu et al., 2025), SpatialVLA (Qu et al., 2025), $\pi_0$ (Black et al., 2024), CogAct (Li et al., 2024) and Villa-x (Chen et al., 2025). On Calvin, we compare UniCoD against several policies that leverage visual generation tasks, including GR-1 (Wu et al., 2023), $\pi_0$ (Black et al., 2024), VPP (Hu et al., 2024), and UP-VLA (Zhang et al., 2025). To ensure a fair comparison, we reproduce these baselines and standardize their visual input to a single third-person view. For $\pi_0$, we specifically use the implementation from the open-pi-zero and report its performance under the same training and evaluation setup used in UniCoD for a direct comparison.

### 4.2.1   PERFORMANCE ON SIMULATION BENCHMARKS

Table 1: Results on SimplerEnv-WindowsX (visual matching). Entries marked with * are methods reproduced with our training and test settings.

| Model | Carrot on Plate | | Eggplant in Basket | | Spoon on Towel | | Stack Cube | | Success |
|---|---|---|---|---|---|---|---|---|---|
| | **Grasp** | **Success** | **Grasp** | **Success** | **Grasp** | **Success** | **Grasp** | **Success** | **Average** |
| RT-1-X | 20.8 | 4.2 | 0.0 | 0.0 | 16.7 | 0.0 | 8.3 | 0.0 | 1.1 |
| Octo-Base | 52.8 | 8.3 | 66.7 | 43.1 | 34.7 | 12.5 | 31.9 | 0.0 | 16.0 |
| OpenVLA | 33.3 | 0.0 | 8.3 | 4.1 | 4.1 | 0.0 | 12.5 | 0.0 | 1.0 |
| RoboVLMs | 33.3 | 20.8 | 91.7 | 79.2 | 70.8 | 45.8 | 54.2 | 4.2 | 37.5 |
| SpatialVLA | 29.2 | 25.0 | 100.0 | **100.0** | 20.8 | 16.7 | 62.5 | 29.2 | 42.7 |
| $\pi_0$* | 58.5 | 48.8 | 78.8 | 64.6 | 83.3 | 73.3 | 62.5 | 12.5 | 49.8 |
| CogAct | / | 58.3 | / | 45.8 | / | 29.2 | / | **95.8** | 57.3 |
| Villa-x | / | 46.3 | / | 64.6 | / | 77.9 | / | 61.3 | 62.5 |
| UniCoD (Ours) | 75.0 | **63.0** | 100.0 | 89.6 | 83.3 | **78.8** | 91.7 | 52.5 | **71.0** |

Tables 1 and 3 present the performance of our method on the SimplerEnv-WindowX and SimplerEnv-Google Robot benchmarks, respectively. We report the officially published results of other methods for comparison. On both robotic platforms, our method achieves the highest success rates of 71.0% and 78.4%, attaining state-of-the-art (SOTA) performance. We highlight the top-performing and second-best methods for each task category in **bold** and with an underline. It is evident that UniCoD demonstrates consistently high success rates across all sub-tasks. This contrasts with other methods, which often exhibit *"spiky"* performance profiles—excelling on some tasks while performing poorly on others. This finding underscores the superior multi-task learning capabilities of our approach.

Furthermore, for a fair, apple-to-apple comparison with the architecturally similar $\pi_0$ baseline, we reproduced it within our identical training and evaluation framework. Across both environments, we found that the novel components in UniCoD yield a significant performance uplift of over 20%. We also observed that this improvement is consistently present at every training checkpoint, indicating that the stable gains can be attributed to our method's ability to learn continuous future features and discrete representations simultaneously.

We also compare UniCoD against several policies that leverage advanced vision-based training methodologies on the Calvin ABC-D split, with results shown in Table 2. Since many prior works utilize multi-view images and historical information, we re-implemented these baselines using a standardized single, third-person-view image as visual input to ensure a fair comparison of the benefits conferred by our training method. The results demonstrate that UniCoD achieves the best performance on single-view manipulation tasks within the Calvin benchmark. Moreover, when compared to the baseline pi0, our method again exhibits a performance improvement, consistent with the results on SimplerEnv.

Table 2: Long-horizon evaluation on the Calvin ABC→D benchmark. Entries marked with * are methods reproduced with our training and test settings. We *only use a single 224x224 third-view image* as input in all methods.

| Method | Tasks completed in a row | | | | | Avg. Len ↑ |
|---|---|---|---|---|---|---|
| | 1 | 2 | 3 | 4 | 5 | |
| RT-1* | 0.533 | 0.222 | 0.094 | 0.038 | 0.013 | 0.900 |
| GR-1 | 0.854 | 0.712 | 0.596 | 0.497 | 0.401 | 3.06 |
| $\pi_0$* | 0.937 | 0.832 | 0.740 | 0.629 | 0.510 | 3.65 |
| VPP* | 0.909 | 0.815 | 0.713 | 0.620 | 0.518 | 3.58 |
| UP-VLA* | 0.928 | 0.865 | 0.815 | 0.769 | 0.699 | 4.08 |
| UniCoD (Ours) | 0.973 | 0.895 | 0.823 | 0.752 | 0.670 | 4.11 |

### 4.3 REAL WORLD EXPERIMENTS

**Implementation Details** We fine-tune the pre-trained UniCoD model separately on the datasets collected from our two real-world robotic platforms to evaluate its performance on a variety of seen and unseen tasks. The fine-tuning process is conducted for 10 epochs using a batch size of 1024 and a learning rate of $5 \times 10^{-5}$, with both the prediction horizon and action chunk length set to 10. For the Franka Emika Panda arm, the model is fine-tuned on 2,000 trajectories, and during deployment, we evaluate both full and half action

Table 3: Results on SimplerEnv-Google Robot (visual matching). Entries marked with * are methods reproduced with our training and test settings.

| Model | Pick Coke | Move Near | O./C. Drawer | Put in Drawer | AVG↑ |
|---|---|---|---|---|---|
| RT-1-X | 56.7 | 31.7 | 59.7 | 21.3 | 42.4 |
| Octo-Base | 17.0 | 4.2 | 22.7 | 0.0 | 11.0 |
| OpenVLA | 16.3 | 46.2 | 35.6 | 0.0 | 24.5 |
| RoboVLMs | 77.3 | 61.7 | 43.5 | 24.1 | 51.7 |
| $\pi_0$* | 93.3 | 78.1 | 23.6 | 12.5 | 51.9 |
| CogACT | 91.3 | **85.0** | **71.8** | 50.9 | 74.8 |
| Villa-x | **98.7** | 75.0 | 59.3 | 5.6 | 59.6 |
| UniCod (Ours) | **98.7** | 81.5 | 63.2 | 70.0 | **78.4** |

chunk execution, reporting the superior result. On the XArm with a 12-DOF dexterous hand, we use a larger dataset of 4,000 trajectories and execute the full 10-step action chunk at each inference step. We test on seen tasks, which involve familiar objects in novel, randomized positions, and unseen tasks, which introduce novel color, objects, and background. For each task configuration, we conduct 20 trials from randomized initial configurations and report the average task success rate. More details can be found in Appendix A.3.

### 4.3.1 PERFORMANCE ON REAL WORLD EXPERIMENTS

We compare UniCoD against OpenVLA(Kim et al., 2024a), GR-1 (Wu et al., 2023), $\pi_0$ (Black et al., 2024), UP-VLA (Zhang et al., 2025) and VPP (Hu et al., 2024) in two environments, visualizing the results in Figure 4 and 5. Our method achieves the highest overall task success rates on both real-world robotic platforms. Specifically, on the Franka Panda arm, UniCoD attains the best performance across all four task categories, outperforming baselines on both seen and unseen tasks. This demonstrates that our approach effectively enhances both multi-task learning and generalization capabilities. Consistent with our findings in the Simpler simulation environment, our method again shows superior performance over the architecturally similar $\pi_0$ baseline across a majority of these real-world tasks. Furthermore, on the more complex 12-DoF dexterous hand platform, UniCoD achieves the highest average success rate across all nine skill categories. Notably, we observe that our method exhibits a significant generalization advantage when dealing with novel objects and scenes.

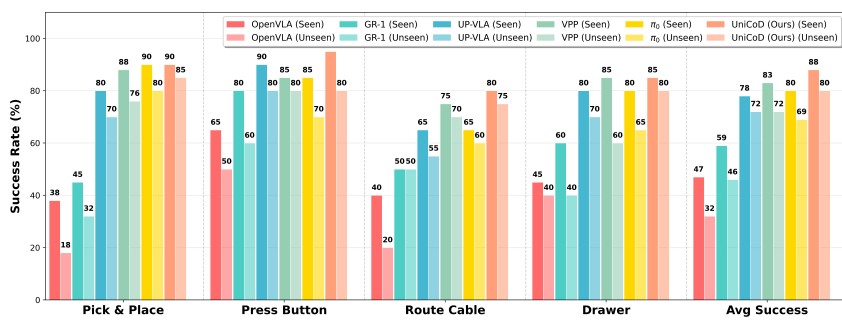

Figure 4: Results on real-world 7DOF robotarm experiment. More detailed quantitative results are provided in Table 6.

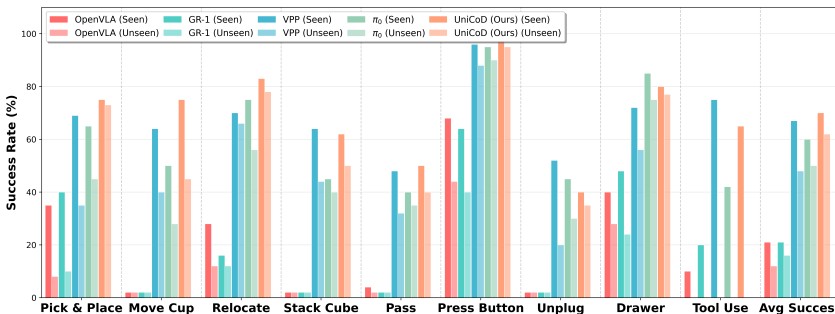

Figure 5: Results on real-world 12-DOF dexterous hands experiment. More detailed quantitative results can be found in Table 7.

We provide several illustrative examples in Appendix A.4, where the model successfully grasps completely unseen objects and correctly interprets out-of-distribution (OOD) language descriptions.

These consistent, state-of-the-art results across two morphologically distinct robots validate the effectiveness and broad applicability of our proposed method.

## 4.4 ABLATION STUDY

In this section, we conduct a series of ablation studies to validate the effectiveness of the different components within UniCoD. These experiments investigate the role of our continuous visual representations, the impact of our large-scale pre-training phase involving both language and visual prediction, and a comparison of several continuous vision encoding methods proposed in Sec 3. All ablation studies are conducted in the Simpler simulation environment, following the same training and evaluation protocols described in Sec 4.2.

**Effectiveness of Continuous Predictive Visual Representations** To validate the effectiveness of prediction using continuous representations, we compare a version of UniCoD without pre-training against two baselines, as shown in Table 4. We evaluate the following without using pretraining: (1) `w/o Continuous` ($\pi_0$), where the modules for predicting continuous future features (including the auxiliary prediction expert and its corresponding encoder/decoder) are removed. (2) `w/Pred`, which predicts future raw pixels using a two-layer MLP. This helps us elucidate the trade-offs between using high-level visual features versus raw pixels

Table 4: Ablation study on unified pretraining paradigm and continuous feature for prediction.

| Model | Carrot | Eggplant | Spoon | Cube | AVG↑ |
|---|---|---|---|---|---|
| w/o Pretrain | | | | | |
| w/o Continuous | 48.8 | 64.6 | 73.3 | 12.5 | 49.8 |
| w/o Continuous w/ Pred | 52.5 | 79.2 | 79.6 | 30.0 | 60.3 |
| UniCoD | 60.8 | 87.1 | 78.8 | 50.4 | 69.3 |
| w/ Pretrain | | | | | |
| UniCoD (Ours) | **63.0** | **89.6** | **78.8** | **52.5** | **71.0** |

as the predictive signal. The results in `w/o Pretrain` section of the table show that our proposed continuous visual feature prediction boosts performance by approximately 20%. Furthermore, the comparison with `w/Pred` reveals that continuous features are indeed a more effective signal for future prediction, enabling the model to extract dynamic information crucial for action generation.

**Effectiveness of Large-Scale Planning and Prediction Pre-training**  Table 4 also presents a comparison between UniCoD with and without pre-training. Overall, pre-training improves the success rate across all tasks, yielding a performance gain of approximately 2%. During fine-tuning, we observe that leveraging large-scale external data for future and language prediction accelerates the model's convergence on the robotics dataset. This effect is particularly pronounced in the convergence of the future prediction loss. This indicates that our joint pre-training scheme, which combines continuous and discrete prediction, provides a superior model initialization, especially for the prediction expert module, which translates to tangible benefits during downstream fine-tuning.

| Method | Google robot | | | | | WidowX robot | | | | |
|---|---|---|---|---|---|---|---|---|---|---|
| | Pick | Move | Drawer | Put | **AVG** | Carrot | Eggplant | Spoon | Cube | **AVG** |
| UniCoD-Distill | 97.2 | 82.6 | **61.9** | **74.4** | **79.0** | 48.8 | **95.8** | **89.6** | 34.6 | 67.2 |
| UniCoD-Dino | **98.3** | **80.2** | 51.1 | 63.3 | 73.2 | 54.6 | 81.7 | 78.8 | 49.6 | 66.1 |
| UniCoD-Siglip | 97.7 | 80.2 | 61.3 | 72.4 | 77.9 | **60.8** | 87.1 | 78.8 | **50.4** | **69.3** |

Table 5: Ablation study on choice of continuous vision features.

**Choice of Continuous Visual Prediction**  We further compare the different encoding methods for future prediction proposed in our methodology. Specifically, we evaluate three distinct approaches (all without pre-training), with results on both Simpler environments shown in Table 5: (1) UniCoD-Distill, which takes the input embeddings of the ViT (from the current frame) as input to the prediction expert and predicts the output features of ViT for the future frame. This approach is analogous to distilling knowledge from the ViT encoder itself. (2) UniCoD-Dino and (3) UniCoD-Siglip, which take the output features of their respective vision encoders (DINO Siméoni et al. (2025) or SigLIP Tschannen et al. (2025)) for the current frame as input to predict the corresponding features for the future frame. The results show that UniCoD-Siglip demonstrates better performance on both benchmarks, and consequently, we select SigLIP as the vision encoder for our UniCoD model. Notably, on Google Robot environment, UniCoD-Distill achieves better performance than the UniCoD-Siglip when neither is pre-trained. This suggests that the distillation-style architecture has inherent advantages. In contrast, UniCoD-Dino performs significantly worse than the other two. This is likely because the DINO feature space is not aligned with the VLM backbone. Conversely, since SigLIP is the native vision encoder for Paligemma, its feature space is naturally more aligned with that of the VLM expert, facilitating more effective integration within the prediction expert.

## 5 CONCLUSION

In this paper, we introduce **UniCoD**, a Vision-Language-Action (VLA) framework that enhances policy learning by integrating discrete token prediction with continuous visual prediction. During the pre-training stage, we leverage embodied VQA and robotic planning tasks to align the discrete language features of a Vision-Language Model (VLM). Concurrently, we train a predictive module on large-scale video data to forecast future continuous visual features. These two components—the VLM backbone and the prediction module—are effectively fused using a Mixture-of-Experts (MoE) Transformer architecture. In the subsequent action fine-tuning stage, an action expert is incorporated, and the entire model is fine-tuned on a joint objective of continuous action generation and future feature prediction. Our method achieves state-of-the-art (SOTA) performance in two distinct simulation environments. Furthermore, on real-world hardware, including a 7-DoF robot arm and a 12-DoF dexterous hand, our model demonstrates superior performance and stronger semantic generalization, particularly when handling novel objects not encountered during training.

# 6 ETHICS STATEMENT

The data used in the VLM4VLA is sourced exclusively from public repositories. Our contributions fully adhere to the terms of these licenses. We did not use any data beyond what is publicly available and downloadable.

# 7 REPRODUCIBILITY STATEMENT

All results and experimental conclusions in this paper are reproducible. To facilitate reproducibility, we have included our source code in the supplementary material. We are committed to open science and plan to publicly release the complete codebase, trained models, datasets, and evaluation logs upon publication of this work.

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

# A  APPENDIX

## A.1  QUALITATIVE COMPARISON OF ENCODED FUTURE VISUAL REPRESENTATIONS

To qualitatively analyze the characteristics of different encoding methods, we visualize the features they produce. Specifically, we compare features from a single robot trajectory encoded in three ways: raw image pixels, continuous visual features from a ViT encoder, and discrete visual tokens from a VQ-GAN. We selected a trajectory from the `Fractal` dataset corresponding to the instruction *pick the coffee bag from the drawer onto the table*. For each frame, the resulting features—raw pixels (flattened from $224 \times 224 \times 3$), ViT features (flattened from $256 \times 1152$), and VQ-VAE tokens (2048-dim)—are first reduced to 50 dimensions via PCA and then projected into a 2D space using t-SNE for visualization.

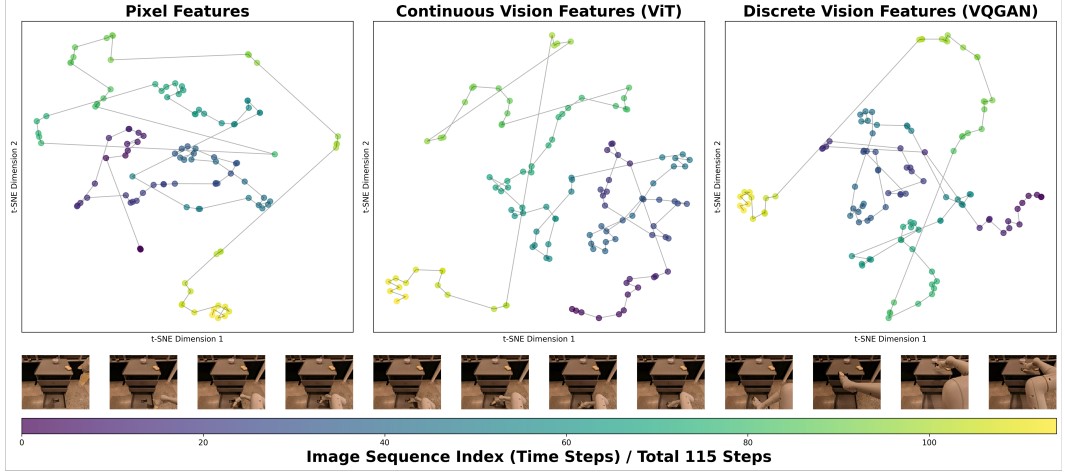

Figure 6: t-SNE Visualization of Different Future Representations.

Figure 6 illustrates the t-SNE visualizations for the trajectory encoded by these three methods. To highlight the temporal evolution, feature points from adjacent frames are connected by lines.

- **Pixel Features (Left):** This encoding preserves the most low-level information. We observe that despite small visual changes between consecutive frames, the corresponding pixel-level features exhibit high variance, often jumping into regions occupied by features from distant timesteps. This suggests that using raw pixel values as a predictive signal could mislead the policy by causing it to over-emphasize low-level, high-frequency changes.
- **ViT vs. VQ Features (Center and Right):** A comparison reveals a distinct *"circling phenomenon"* in the VQ-GAN visualization, where features from many different timesteps collapse into a dense central region. This indicates poor temporal separability in the context of manipulation trajectories. In contrast, the ViT features provide the best separation of the three methods, organizing features from different frames into distinct, minimally overlapping clusters.

This qualitative analysis supports our insight that continuous features, by virtue of focusing on high-level semantic information, serve as a more stable and suitable predictive signal for robot action policies within our framework.

## A.2  DETAILS ABOUT SIMULATION BENCHMARKS

**Calvin Benchmark**  Calvin is a simulation benchmark designed for evaluating long-horizon, language-conditioned manipulation policies. It comprises four distinct environments (A, B, C, and D) and offers evaluation splits such as *ABC-D* and *ABCD-D*. In our experiments, we employ the *ABC-D* split to evaluate the single-view generalization capabilities of the models. Models are trained on data collected from environments A, B, and C, and subsequently evaluated in the unseen

environment D. This evaluation suite includes 34 different manipulation tasks organized into 1,000 long-horizon sequences, each of length 5. We report the average length of successfully completed sub-task sequences.

**SimplerEnv Benchmark**    SimplerEnv is a simulation benchmark designed to evaluate policies trained on large-scale real-world datasets, such as Bridge-V2 and Fractal. It procedurally generates scenes that mimic real-world environments using texturing techniques, allowing models trained on real data to be tested directly in simulation without requiring physical deployment. The benchmark supports two types of robot arms: the WindowX and the Google Robot. For our evaluation, we conduct 240 runs for each task and report the average success rate.

### A.3   DETAILS ON REAL WORLD EXPERIMENTS

### A.3.1   FRANKA PANDA ROBOT ARM

**Real-World Franka Emika Panda Arm**    We deploy several models on a Franka Emika Panda arm for real-world task comparison. The robot arm features 7 degrees of freedom (DoF). Its action space is defined by a 7-dimensional vector, where the first six dimensions specify the relative change in the end-effector's 6D pose (3D position and 3D orientation), and the final dimension controls the binary state of the gripper (open or closed). In our experiments, the policy takes images from an on-board, first-person-view camera as visual input and outputs these relative actions. We first collected a dataset of 2,000 trajectories spanning over 20 distinct tasks, encompassing six fundamental skills: picking, placing, opening a drawer, closing a drawer, pressing a button, and routing a cable. We evaluate performance on both seen and unseen task variations. The unseen category primarily involves grasping novel objects not present in the training data.

The task suite for the Franka Panda arm includes:

- **Pick & Place:** Grasping and placing a variety of objects. The training set includes items such as a toy banana, a toy eggplant, red/green/blue blocks, and red/yellow/black plates.

- **Press Button:** Pressing a toy button using a grasped black block as a tool.

- **Route Cable:** Routing a thin black rubber cable into a narrow slot.

- **Drawer Operation:** Opening a toy drawer.

**Unseen Tasks**    These are designed to evaluate generalization: *Novel Objects:* Grasping objects not seen during training (e.g., toy chili, toy strawberry, yellow block, large toy eggplant, arrow sticker, marker pen). *Distractors:* Operating in the presence of irrelevant distractor objects. *Visual Variations:* Adapting to changes in background color and object color.

We tested UniCoD, OpenVLA(Kim et al., 2024a), GR-1(Wu et al., 2023), $\pi_0$ (Black et al., 2024), UP-VLA (Zhang et al., 2025) and VPP (Hu et al., 2024) on this environment. The detailed results are shown in Table 6 (corresponding to Figure 4).

Table 6: Detailed results on Franka-Emika Panda Robotarm. We evaluate each task 20 times (100 trials per skill) with random initialization and report the average success rate.

| Model | Pick & Place | | Press Button | | Route Cable | | Drawer | | Avg Success | |
|---|---|---|---|---|---|---|---|---|---|---|
| | Seen | Unseen | Seen | Unseen | Seen | Unseen | Seen | Unseen | **Seen** | **Unseen** |
| OpenVLA | 38 | 18 | 65 | 50 | 40 | 20 | 45 | 40 | 47 | 32 |
| GR-1 | 45 | 32 | 80 | 60 | 50 | 50 | 60 | 40 | 59 | 46 |
| UP-VLA | 80 | 70 | 90 | 80 | 65 | 55 | 80 | 70 | 78 | 72 |
| VPP | 88 | 76 | 85 | 80 | 75 | 70 | 85 | 60 | 83 | 72 |
| $\pi_0$ | 90 | 80 | 85 | 70 | 65 | 60 | 80 | 65 | 80 | 69 |
| UniCoD (Ours) | **90** | **85** | **95** | **80** | **80** | **75** | **85** | **80** | **88** | **80** |

### A.3.2 XArm Dexterous Manipulation

**Real-World XArm with 12-DOF X-Hand**  Our 12-DoF single-arm dexterous manipulation platform, which comprises a 7-DoF XArm and a 5-DoF hand, is controlled using a dual-view visual input from both first-person and third-person cameras. During evaluation, we test pick-and-place capabilities across 5 distinct task variations for a total of 50 trials. For all other skills, we conduct 20 trials per task. The final performance is reported as the average success rate for each skill. We train different models using a dataset of 4,000 trajectories across more than 100 tasks. The models are then evaluated in a variety of seen and unseen scenarios, which cover 13 distinct skills, e.g., picking, placing, stacking, and pouring. To specifically test for visual generalization, we alter the background colors and novel objects during evaluation in the unseen scenarios.

The task suite for the XArm platform includes:

- **Dexterous Pick & Place:** Dexterously grasping and placing a wide range of objects. The training set includes a toy banana, a toy eggplant, a toy orange, small and large toy soccer balls, a computer mouse, a toy drawer, and more.
- **Move Cup:** Grasping and moving a cup to a different location.
- **Relocate:** Grasping an object and placing it adjacent to another target object.
- **Stack Cube:** Placing one block on top of another.
- **Pass:** Grasping an object and handing it to a human operator.
- **Press Button:** Directly actuating a toy button with a finger.
- **Unplug:** Extracting a rubber cable from a socket.
- **Drawer Operation:** Opening or closing a toy drawer.
- **Tool Use:** Using various tools, such as a spoon (e.g., for scooping) and a toy hammer (e.g., for striking).

**Unseen Tasks**  These are designed to evaluate generalization: *Novel Objects:* Grasping unseen objects and placing them to not-seen targets during training (e.g., apple, lemon, glass cup, glass plate, blue plate, toy kapibla, transparent plate, green apple, big ball, and various of novel objects). *Distractors:* Operating in the presence of irrelevant distractor objects. *Visual Variations:* Adapting to changes in background color and object color.

Table 7: Detailed results on XArm with dexterous hand. We evaluate 50 times on Pick & Place tasks and 20 trials on other tasks with random initialization and report the average success rate.

| Model | Pick & Place | | Move Cup | | Relocate | | Stack Cube | | Pass | |
|---|---|---|---|---|---|---|---|---|---|---|
| | Seen | Unseen | Seen | Unseen | Seen | Unseen | Seen | Unseen | Seen | Unseen |
| OpenVLA | 35 | 8 | 0 | 0 | 28 | 12 | 0 | 0 | 4 | 0 |
| GR-1 | 40 | 10 | 0 | 0 | 16 | 12 | 0 | 0 | 0 | 0 |
| VPP | 69 | 35 | 64 | 40 | 70 | 66 | **64** | 44 | 48 | 32 |
| $\pi_0$ | 65 | 45 | 50 | 28 | 75 | 56 | 45 | 40 | 40 | 35 |
| UniCoD (Ours) | **75** | **73** | **75** | **45** | **83** | **78** | 62 | **50** | **50** | **40** |

| Model | Press Button | | Unplug | | Drawer | | Tool Use | | Avg Success | |
|---|---|---|---|---|---|---|---|---|---|---|
| | Seen | Unseen | Seen | Unseen | Seen | Unseen | Seen | Unseen | **Seen** | **Unseen** |
| OpenVLA | 68 | 44 | 0 | 0 | 40 | 28 | 10 | / | 21 | 12 |
| GR-1 | 64 | 40 | 0 | 0 | 48 | 24 | 20 | / | 21 | 16 |
| VPP | 96 | 88 | **52** | 20 | 72 | 56 | **75** | / | 67 | 48 |
| $\pi_0$ | 95 | 90 | 45 | 30 | **85** | 75 | 42 | / | 60 | 50 |
| UniCoD (Ours) | **97** | **95** | 40 | **35** | 80 | **77** | 65 | / | **70** | **62** |

### A.4 Examples of Demos on OOD-Tasks

Examples of video on unseen objects are shown in Figure 7, where unseen objects are bold in the instructions. More demos can be found in our anonymous website.

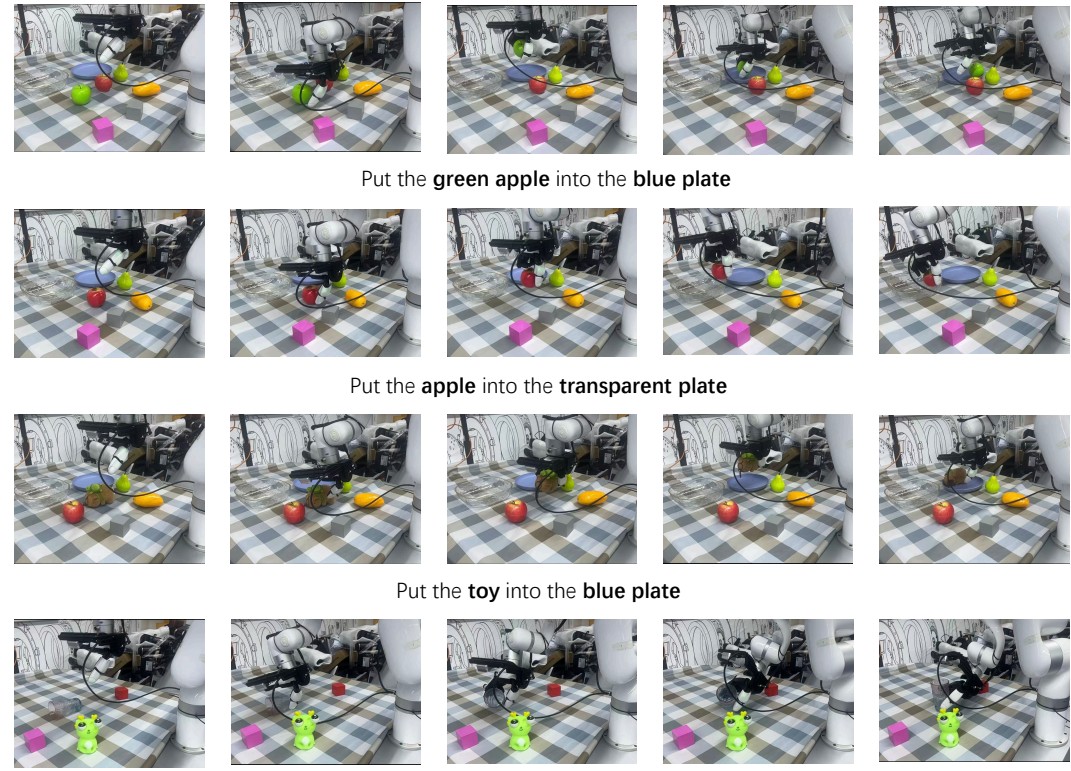

Put the **green apple** into the **blue plate**

Put the **apple** into the **transparent plate**

Put the **toy** into the **blue plate**

Lift up the **transparent cup**

Figure 7: Examples of Semantic Generalization to OOD objects

## A.5 DATA USED FOR PRE-TRAINING

Table 8 summarizes the datasets employed during pre-training. To creating the robot vqa data, we employ Gemini 2.5(Comanici et al., 2025) to annotate text descriptions and task planning for a subset of video data. The RoboMind dataset inherently contains overall task descriptions and sub-tasks, which can be directly utilized as vision–language question–answer pairs.

Table 8: Datasets and the number of samples used for TI2E task and VQA task.

| Task name | Dataset name | Number of samples |
|---|---|---|
| **TI2E** | AgibotWorld(Bu et al., 2025) | 120k |
| | Galaxea Open-World(Jiang et al., 2025) | 99k |
| | Robomind(Wu et al., 2024) | 20k |
| | Droid(Khazatsky et al., 2024) | 76k |
| | Bridge(Walke et al., 2023) | 55k |
| | Egodex(Hoque et al., 2025) | 320k |
| | Ego4D(Grauman et al., 2022) | 500k |
| **VQA** | AgibotWorld VQA | 120k |
| | Galaxea Open-World VQA | 99k |
| | Robomind VQA | 20k |
| | Droid VQA | 76k |
| | LLaVA-Pretrain(Liu et al., 2023) | 558k |

## A.6 VQA DATA DESIGH

We present several examples of embodied VQA question–answer pairs in Figure 8.

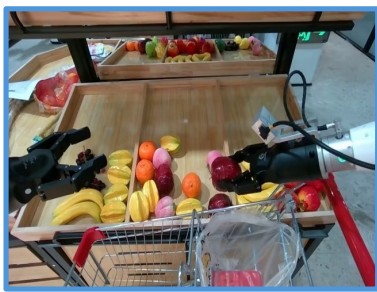

Question: Based on current observations and instruction < Put apples, oranges and peaches in plastic bags.>, what subtask does the robot need to do next?

Answer: Next subtask is: place the held red apple into the plastic bag in the shopping cart.

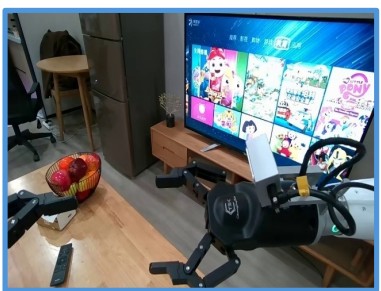

Question: Based on current observations and instruction < Using the TV Remote.>, what subtask does the robot need to do next?

Answer: Next subtask is: Pick up the remote control from the table with left arm.

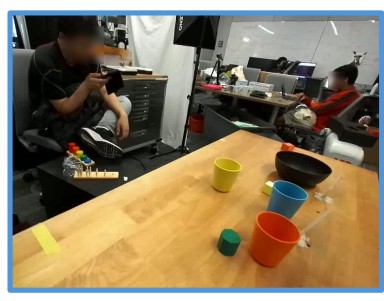

Question: Based on current observations and instruction < Stacking the blue, yellow, and orange cups.>, planning for the task the robot need to do?

Answer: Planning: Pick the blue cup; Place the blue cup on the yellow cup; Pick up the yellow cup; Place the yellow cup on the orange cup

Figure 8: Example of VQA.

For part of the embodied datasets (e.g., Agibot and RoboMIND), which contain precise instruction descriptions, we can directly construct QA pairs. For other datasets, we employ Gemini to decompose and annotate instruction descriptions according to the following prompt in Figure 9, 10.

## B USAGE OF LLMS

In the final stages of preparing this manuscript, the authors used a Large Language Model (LLM) solely for grammar checking and language polishing. The model assisted in improving sentence structure and correcting grammatical errors to enhance readability.

**Prompt Example**

**System Message:**

```
"You are an expert in video analysis and robotic task
    understanding.
You will be given an image sequence representing a video and a
    reference
description. Your task is to decompose the total task into
    several steps
which are needed to complete the task, and label each step with a
    frame range."
```

**User Message:**

```
## Task Description
You will analyze an image sequence of a robotic arm performing a
    specific task.
Your task is to make the overall task description more detailed
    with the help of
the video clip, extract the necessary steps, and specify the
    frame range for each step.

## Target
Step Extraction: Extract the key steps required to complete the
    task. Each step includes:
- Specific actions decomposed from video and description
- Frame window: Specify the start and end frame for each step

## Requirements:
1. Different steps must correspond to different action types.
2. A step cannot contain two or more actions.
3. Two similar steps need to be merged into one.
4. The first step must start at frame 0, and the last step cannot
    exceed {frame_num-1}.

## Output Format
Return output in JSON:
{
  "task_summary": "...",
  "steps": [
    {"step_description": "...", "start_frame": 0, "end_frame": 6},
    {"step_description": "...", "start_frame": 7, "end_frame": 12}
  ]
}

## Example Input
"task description": "Moving colored blocks into a container."
"video": image sequence with length {frame_num}
```

1

Figure 9: prompt for Gemini.

```
## Example Output
{
  "task_summary": "Moving the red and yellow blocks into a
      container.",
  "steps": [
    {"step_description": "pick the red block.", "start_frame": 0,
        "end_frame": 6},
    {"step_description": "place the red block into container.",
        "start_frame": 7, "end_frame": 12},
    {"step_description": "pick the yellow block.", "start_frame":
        13, "end_frame": 15},
    {"step_description": "place the yellow block into
        container.", "start_frame": 16, "end_frame":
        {frame_num-1}}
  ]
}
```

2

Figure 10: prompt for Gemini.