# OpenReview forum: "UniCoD: Enhancing Robot Policy via Unified Continuous and Discrete Representation Learning"
_ICLR.cc/2026/Conference — ICLR 2026 Conference Withdrawn Submission_

### Official Review · Reviewer_TXuu · 2025-10-23

**Soundness:** 2
**Presentation:** 2
**Contribution:** 2
**Rating:** 4
**Confidence:** 4

**Summary:**

The authors propose a VLA called UniCoD. The model is based on the mixture of transformers paradigm trained on both continuous and discrete losses. The model is trained in two stages: the first stage only contains vision-language data and trains the model on CE next-token prediction and  MSE loss. The second stage introduces action data and trains a flow-matching expert, as well as the MSE loss for the generation expert. The authors present good results on SimplerEnv-WindowsX [table 1], calvin [table 2] and on a real-world robot [figures 4,5].

**Strengths:**

- The authors provide gains on multiple benchmarks and real-world evaluations.
- The ablations in Table 4 clearly show that the MSE loss helps.

**Weaknesses:**

1. The novelty is limited.
2.  The used baselines are not consistent among tables (e.g, table 2 doesn't have octo.) and standard baselines like Groot models are missing.
3. The writing is not very clear about what the contribution is.

**Questions:**

1. Why are the baselines not consistent among tables? E.g., table 2 doesn't have octo. Can you add all baselines to all tables?
2. Table 4 seems to show that pretraining barely helps, it only gives 2 points. Is that correct?
3. Can you clarify what the novelty is?
4. Could you add error bars to your results?
5. Could you ablate the use of discrete predictions too in table 4?

---

> ### Author Response · Authors · 2025-11-21
> **Rebuttal to Review TXuu (Part1)**
>
> We sincerely appreciate your time and efforts in reviewing our paper! Based on your review, we added a detailed discussion.
>
> ---
>
> **1: About novelty and our contribution (W1, W3, Q3)**
>
> Thank you for highlighting this issue. We agree that the description of our contribution and novelty in the original submission may have been insufficiently precise. We clarify our contributions as follows:
>
> (1) Novel joint training framework for VLA.
> We introduce a new joint training framework specifically designed for VLA tasks, where embodied language understanding and dynamics prediction (expressed through continuous visual representations rich in semantics) are trained simultaneously. This design is fundamentally different from prior work, which typically focuses on pixel-level prediction and either neglects high-dimensional visual representation learning or does not leverage robot-domain data to maintain the pretrained multimodal understanding and planning capabilities of VLMs.
>
> (2) First use of Mixture-of-Transformers for three heterogeneous tasks.
> Our method is, to our knowledge, the first to employ a Mixture-of-Transformers architecture to jointly address three tasks: understanding (planning), dynamics modeling, and action prediction. Compared to methods such as UP-VLA and CoT-VLA, our framework better preserves the pretrained capabilities while enabling effective task-specific adaptation.
>
> (3) Analysis of visual-representation prediction for VLA performance.
> We further provide a systematic discussion on how learning different forms of visual-representation prediction influences VLA performance. This aspect has not been examined in prior works and forms a key motivation behind our framework design.
>
> ---
>
> **2: About inconsistent baselines among tables? (W2, Q2)**
>
> We greatly appreciate your suggestion. In the final version of the paper, we will supplement our experiments by reproducing the results of all baselines across the benchmarks.
> We would also like to provide some context regarding the initial inconsistency in baseline selection: several older baselines did not officially evaluate or report results on these specific benchmarks. Due to time and computational resource constraints, we were previously unable to fully re-implement and test every method across all simulations. However, we are now working to close this gap to ensure a comprehensive comparison. Here, we provide several missing baselines that are reproduced under our settings in Table 2. (Villa-x has not been open-sourced yet so we didn't include its results on Calvin.)
>
> |              | 1     | 2     | 3     | 4     | 5     | All  |
> |--------------|-------|-------|-------|-------|-------|------|
> | Octo-Base*   | 0.312 | 0.141 | 0.072 | 0.042 | 0.018 | 0.59 |
> | OpenVLA*     | 0.726 | 0.425 | 0.235 | 0.132 | 0.081 | 1.60 |
> | SpatialVLA*  | 0.942 | 0.841 | 0.749 | 0.628 | 0.504 | 3.66    |
> | RoboVLMs*    | 0.878 | 0.721 | 0.591 | 0.498 | 0.408 | 3.10 |
> | CogAct*      | 0.883 | 0.763 | 0.658 | 0.573 | 0.483 | 3.36 |
> | Ours(UniCoD) | 0.973 | 0.895 | 0.823 | 0.752 | 0.670 | 4.11 |

---

> > ### Author Response · Authors · 2025-11-21
> > **Rebuttal to Review TXuu (Part2)**
> >
> > ---
> >
> > **3: Table 4 seems to show that pretraining barely helps, it only gives 2 points. Is that correct? (Q3)**
> >
> > Since the pre-training exclusively uses real-world scenario data, the improvement for simulation tasks is not significant—this is because the image style and scenarios rendered in simulation differ greatly from those in reality, and there are only a few single in-distribution objects. Consequently, the performance advantage cannot be intuitively reflected. However, the advantages of pre-training can be observed through comparing the results of some unseen tasks in real-robot experiments, as shown in the following table:
> >
> > | Approach        | place the apple on the transparent plate   | place the lemon on the basket|
> > |:---------------:|:-------------------:|:-------------------:|
> > | Unicod              | 14/20      | 15/20     |
> > | Unicod(w/o pre-training)        |  9/20           |  11/20 |
> > | Unicod(w/o Discrete-Pretrain)        |  14/20           |  12/20 |
> >
> > Here, both the transparent plate and the lemon are unseen objects during training, and there was only one object placement position in the training process. Models without pre-training tend to randomly select plates for placement, and the accuracy of grasping lemons decreases. Therefore, pre-training is highly effective in real-world manipulation tasks, particularly for the generalization to unseen tasks.
> >
> > ---
> >
> > **4: Could you ablate the use of discrete predictions too in table 4? (Q5)**
> >
> > Thank you for the suggestion. In pretraining, we used both discrete prediction and continuous visual feature prediction; during action fine-tuning, we used continuous visual feature prediction together with action learning. To ablate the effect of discrete predictions, we modified the pretraining stage to use only continuous visual features as the training objective, while keeping the action fine-tuning stage unchanged. The results are as follows in our answer 3.
> >
> > ---
> >
> > **5: Could you add error bars to your results? (Q4)**
> >
> > Thank you for this suggestion. In the final version of the paper, we will add the standard deviation of success rates obtained from repeated experiments.
> >
> >
> > ---
> > Thank you again for your time and effort in reviewing our work! We hope this clarification can solve your concerns!

---

> > > ### Author Response · Authors · 2025-11-28
> > >
> > > Dear Reviewer TXuu:
> > >
> > > We sincerely appreciate the time and effort you have taken to review our rebuttal. In response to your insightful feedback, we have carefully incorporated the requested baseline comparisons and highlighted the novel training paradigm of our approach. Could we politely ask if there are any further concerns existed? We are always willing to address any of your further questions. If there are no additional concerns, we sincerely wish you could reconsider your score.
> > >
> > > Thank you once again for your valuable time！
> > >
> > > Best Regards,
> > >
> > > The Authors

---

### Official Review · Reviewer_Cpwz · 2025-10-27

**Soundness:** 3
**Presentation:** 3
**Contribution:** 3
**Rating:** 2
**Confidence:** 5

**Summary:**

This paper introduces UniCoD, a novel framework for training generalist robot policies by unifying discrete and continuous representation learning. The core challenge it addresses is that existing methods typically rely on either vision-language models (for understanding) or generative models (for dynamics), while both are crucial for robotics. UniCoD learns to simultaneously understand tasks via discrete language representations and model world dynamics by predicting continuous future visual features. Experiments conducted in simulation (SimplerEnv, Calvin) and on two real-world platforms (a 7-DOF arm and a 12-DOF dexterous hand) show that UniCoD achieves state-of-the-art performance. It significantly outperforms baseline methods, demonstrating superior generalization to novel objects and tasks.

**Strengths:**

The method involves a two-stage training process:
1. Pre-training: The model is first pre-trained on over 1 million internet-scale instructional videos and embodied VQA data to learn these joint representations. It's a heavy work.
2. Fine-tuning: An action expert is then added, and the model is fine-tuned on robot-specific data, learning to map its predictive representations to action tokens.

**Weaknesses:**

1. This framework is followed the understanding&generation framework. Please discuss the difference between CoT-VLA, Up-VLA and HybridVLA.
2. The author need to give more information about the efficiency of the model control, especially when the paper utilize such heavy framework to perform the robot control.
3. I wonder whether the ``pretraining on over 1M internet-scale instructional manipulation videos'' is beneficial for all downstream manipulation tasks. Particularly, please show that the generation quality of the images after the downstream manipulation finetuning.
4. I don't know the meaning of utilizing the such kind of heavy pipeline to perform VLA. The performance improvement is completely unable to offset the overall computational overhead, like compared to CogACT in SimplerEnv-Google Robot Benchmarks.

**Questions:**

See weakness.

---

> ### Author Response · Authors · 2025-11-21
> **Rebuttal to Review Cpwz (Part1)**
>
> We sincerely appreciate your time and efforts in reviewing our paper! Based on your review, we added a detailed discussion and additional experiments.
>
> ---
>
> **1: Discuss the difference between CoT-VLA, Up-VLA and HybridVLA (W1)**
>
> CoT-VLA and Up-VLA both use a unified Transformer network to predict goal images and actions. Our method differs from theirs in the following key aspects:
>
> 1. Compared to pixel-level prediction, we use continuous, pretrained vision-encoder representations as supervision signals.
> Since vision-encoder features contain richer semantic information, our approach shows clear advantages in the ablation study compared with pixel prediction.
>
> 2. Unlike CoT-VLA and Up-VLA, which share KVQ weights across modalities, we adopt MOT to design separate weight networks for each modality while still performing unified attention computation.
> The benefit is that we preserve the original pretrained model’s capabilities while enabling the model to effectively handle additional modalities. Although this introduces new trainable parameters, the parameters in the action expert and generation expert modules account for only a small proportion of the original model (only about 22%), so they do not affect inference speed.
>
> 3. Large-scale Joint Feature Pretraining:
> Unlike CoT-VLA and UP-VLA rely on limited data sources, primarily consisting of subsets from Open-X, Epic-Kitchens, and Something2Something. UniCoD, however, incorporates a much broader and larger-scale dataset, including AgibotWorld, Galaxea Open-World, and Droid, alongside human-hand video datasets such as Ego4D and EgoDex.
> In UniCoD, we leverage a massive corpus of approximately 1 million video clips, encompassing both manipulation tasks and general scenarios. We employ a joint objective combining discrete language task planning and continuous visual feature prediction to pre-train both the VLM and the generation expert. This strategy significantly enhances the model's generalization capabilities towards novel objects in real-world tasks.
>
>
> For HybridVLA, I believe its focus is quite different from ours, so the relevance is limited. HybridVLA mainly concentrates on replacing the traditional VLA action decoder with a combined autoregressive–diffusion action head. It does not involve multi-modal future prediction or the corresponding pretraining components. That said, HybridVLA is still a representative and influential VLA approach, and we will include a citation to it in our paper.
>
>
> ---
>
> **2: Efficiency of the model control? (W2)**
>
> This is a meaningful question. The inference latency of our model on the H20 GPU is shown in the following table (we test 100 times of forward pass and report the average latency).
>
> | Approach        | Latency ms |
> |---------------|-------------------|
> | Unicod(Ours)              | 260      |
> | pi0        |  250            |
>
> Compared with the baseline pi0, our method has almost the same latency. This is because the parameters of the VLM expert are similar. Our proposed UniCOD comprises 2.9B parameters, including a 2.3B VLM, 0.3B generation expert, and 0.3B action expert. Although the additional experts introduce extra parameters, they only increase the pre-training overhead and do not lead to higher inference latency (thanks to token parallel processing and the relatively small size of the generation and action experts).

---

> > ### Author Response · Authors · 2025-11-21
> > **Rebuttal to Review Cpwz (Part2)**
> >
> > ---
> >
> > **3: I wonder whether the ``pretraining on over 1M internet-scale instructional manipulation videos'' is beneficial for all downstream manipulation tasks. Particularly, please show that the generation quality of the images after the downstream manipulation finetuning. (W3)**
> >
> > In fact, pre-training at a certain scale can have varying impacts on different downstream tasks, and some tasks may benefit very little. For example, in simulation scenarios, most pre-training data comes from the real world, which differs from the style of simulation-rendered images. Additionally, simulation scenarios themselves are relatively simple, resulting in limited improvement in simulation performance from pre-training.
> >
> > In contrast, in real-world tasks—such as the "place the apple on the transparent plate" task in the pi0, which was not trained on large amounts of data for visual signal prediction, may fail to distinguish between descriptions of transparent and blue objects (transparent objects were not seen in the real-robot collected data). However, benefiting from the inclusion of such objects as interaction targets in the pre-training data (present in AgiBot), our model can align with this semantics. (Demos and task details can be found in *Appendix A.4* and our anonymous website *https://sites.google.com/view/uni-cod*.)
> >
> > Since our method does not predict image pixels but instead predicts the dynamic representations of the vision encoder, it cannot provide intuitive visualization results.
> >
> > ---
> > **4: I don't know the meaning of utilizing the such kind of heavy pipeline to perform VLA. The performance improvement is completely unable to offset the overall computational overhead, like compared to CogACT in SimplerEnv-Google Robot Benchmarks. (W4)**
> >
> > This is a crucial question. We would first clarify that our proposed architecture is not a computationally heavy pipeline, which allows our inference efficiency to remain on par with smaller VLAs such as pi0, as explained in W2.
> > Our model has a total of 2.9B parameters, consisting of 2.3B for the VLM and 0.3B each for the Generation Expert and Action Expert. In contrast, the CogACT framework you mentioned utilizes a 7B Prismatic VLA, resulting in a significantly larger parameter count. From this perspective, it is worth noting that our method achieves superior performance on the Simpler benchmark while requiring significantly fewer parameters than CogACT.
> >
> > Second, the additional parameters in Unicod primarily serve as a bridge to unify the training of cross-ontology data. Given that the action spaces of cross-ontology data are heterogeneous—while variations in their image features are independent of action space choices—training such an expert network from scratch aims to leverage all video data and avoid information loss caused by action space heterogeneity. Thus, the pipeline with a multi-modal expert and two-stage training strategy is very useful for the VLA foundation model.
> >
> > ---
> >
> > Thank you again for your time and effort in reviewing our work! We hope our clarification can solve all your concerns and shows the improved quality of our paper!

---

> > > ### Author Response · Authors · 2025-11-28
> > >
> > > Dear Reviewer Cpwz:
> > >
> > > We sincerely appreciate the time and effort you have taken to review our rebuttal. In response to your insightful feedback, we have carefully incorporated the requested comparison to other methods and included a discussion of the details of our model. Could we politely ask if there are any further concerns existed? We are always willing to address any of your further questions. If there are no additional concerns, we sincerely wish you could reconsider your score.
> > >
> > > Thank you once again for your valuable time！
> > >
> > > Best Regards,
> > >
> > > The Authors

---

### Official Review · Reviewer_pt64 · 2025-10-31

**Soundness:** 4
**Presentation:** 4
**Contribution:** 4
**Rating:** 8
**Confidence:** 4

**Summary:**

UniCoD is a unified multimodal framework that uses a MoT architecture to integrate text understanding, visual prediction, and action execution for robotic manipulation.

In the first stage, it learns joint vision–language embeddings by aligning textual instructions with visual observations and predicting future visual states in a continuous feature space using a frozen visual encoder. Then, UniCoD fine-tunes this model with embodiment data by introducing an action expert that learns continuous action distributions via flow matching, enabling coherent mapping from multimodal inputs to robot actions.

Results show that UniCod achieves state-of-the-art results across both simulated and real-world environments.

**Strengths:**

By jointly training on text, vision, and continuous future prediction, UniCoD builds deeply aligned multimodal embeddings that are resilient to noise or missing information in any single modality.

The modular MoT design enables selective fine-tuning (e.g., only the action expert), reducing computational cost and preventing catastrophic forgetting of general skills.

The dual-objective training (cross-entropy for language, MSE for vision, flow matching for actions) helps maintain stable convergence and robust multimodal coordination.

**Weaknesses:**

The model design with multiple expert modules (for language, vision, generation, and action) requires substantial computational resources and careful coordination. This can make training costly, difficult to reproduce, and potentially unstable without large-scale infrastructure. Are there analysis on the cost for fine-tuning on new tasks and new envs?

How about the interpretability of the model? For example, are there any safety or hazard-preventing modules or self-correcting modules of the model?

**Questions:**

See weaknesses

---

> ### Author Response · Authors · 2025-11-21
>
> We sincerely appreciate your time and efforts in reviewing our paper! Based on your review, we added a detailed discussion and additional experiments.
>
> ---
>
> **1: Difficult to reproduce, and potentially unstable (W1)**
>
> We agree with the reviewers' point that our pipeline, which includes three distinct expert modules, complicates both the model and its training. In fact, the first-stage pre-training does not involve the action space and only includes the alignment of text and future image features. We did not observe notable instability during training. We experimented with different data mixing ratios and found that the model consistently converged to satisfactory performance across settings. We will make the code and data for this stage publicly available to ensure reproducibility. For the action fine-tuning in the second stage, one can choose to use only the robot's ontology data to fine-tune future feature and action prediction, as the model has already completed pre-training in the robot domain in the first stage. The code and weights for two stages will also be released.
>
> ---
>
> **2: Are there analysis on the cost for fine-tuning on new tasks and new envs?**
>
> Since the model has already learned dynamics prediction from a large amount of data, the cost of fine-tuning it to new environments and tasks is relatively low. For our real robot, we use 25 expert trajectories per task, with a total of 100+ tasks. The fine-tuning process can be carried out using 8xH20/A100 GPUs for 7,000 steps.
>
> ---
>
> **3: How about the interpretability of the model? For example, are there any safety or hazard-preventing modules or self-correcting modules of the model?**
>
> Your question is highly meaningful. Our model does not currently include this part of the training data and modules. However, as you mentioned, this content is important and worthy of research for general embodied intelligence. In fact, our model incorporates a large VLM foundation; in future work, we can integrate this part of the training data to achieve safe and controllable decision-making. Thanks for your valuable advice!
>
> ---
>
> Thank you again for your time and effort in reviewing our work! We hope our clarification can solve all your concerns, and we are always ready to answer any further questions!

---

### Official Review · Reviewer_Nbom · 2025-11-03

**Soundness:** 3
**Presentation:** 4
**Contribution:** 3
**Rating:** 8
**Confidence:** 2

**Summary:**

UniCoD is a VLA policy that couples discrete language/understanding tokens with continuous predictions of future visual features in a Mixture-of-Transformers. The model is pretrained on ~1M instructional/manipulation videos to predict future visual embeddings and VQA/planning tokens, then fine-tuned with an action expert via flow matching while retaining the future-prediction head. The premise is that policy learning improves when semantic reasoning, planning, and explicit future representations are learned jointly. Reported experiments show state-of-the-art results on SimplerEnv and CALVIN, plus strong performance on two real-robot setups. Ablations (Table 4) suggest that future-feature prediction accounts for a substantial share of the gains.

**Strengths:**

1. The proposed procedure is intuitive and well-motivated, combining discrete reasoning with continuous visual forecasting.

2. The authors did conduct extensive experimental analyses, including on two real-world robots, with a clearly detailed protocol. The results show that UniCoD achieves the state-of-the-art performance in multiple benchmarks.

3. The paper is clearly written, and the ablation studies performed by the authors are thorough and convincing.

**Weaknesses:**

1. While the experimental results are detailed, they are only point-estimates with no std or confidence-intervals. The authors should add these, since they did multiple trials. This is a significant weakness.

**Questions:**

1. Please add statistical analyses of the results for multiple trials.
2. The dataset is very large. Is it guaranteed that unseen objects are not in the training dataset ?

---

> ### Author Response · Authors · 2025-11-21
>
> We sincerely appreciate your time and efforts in reviewing our paper! Based on your review, we added a detailed discussion.
>
> ---
>
> **1: About adding std or confidence intervals and statistical analyses of the results for multiple trials (W1, Q1)**
>
> Thank you for this suggestion. In the final version of the paper, we will add the standard deviation of success rates obtained from repeated real-world experiments.
>
>
> ---
>
> **2: The dataset is very large. Is it guaranteed that unseen objects are not in the training dataset? (Q2)**
>
> Thank you for your insightful comment. To clarify, our method utilizes a two-stage data pipeline:
>
> - Pre-training Stage: This stage focuses on learning discrete language representations and continuous space world modeling using 1 million internet-scale instructional manipulation videos. While this dataset covers a vast range of general scenarios—and the model may encounter objects or linguistic concepts similar to those in the test set—crucially, no low-level actions are learned during this phase. Similar to the motivation behind Large Language Models (LLMs), the goal here is to acquire robust semantic generalization capabilities from extensive general-purpose data.
>
> - Action Fine-tuning Stage: This stage uses embodiment-specific trajectory data containing action labels. The dataset size is deliberately kept small (e.g., only 2k self-collected trajectories for the physical Franka-Emika robot and 4k for the 12-DoF dexterous hand). We strictly ensure that these fine-tuning trajectories do not include any unseen objects designed for the Out-of-Distribution (OOD) testing tasks.
>
> By strictly separating action learning from unseen objects, we aim to demonstrate that the model's performance stems from the semantic generalization acquired during pre-training, rather than memorization of specific object interactions. In fact, the unseen objects in our real-world experiments (e.g., unseen fruits, toys, and transparent plates) were items we prepared ourselves. They definitely did not appear in the pre-training data.
>
> ---
>
> We hope our clarifications address your concerns and demonstrate the improved quality of our paper! Please feel free to reach out with any further questions.
> Thank you again for your valuable time!

---

### Author Response · Authors · 2025-12-01

Dear Area Chair and Reviewers,

We sincerely thank all reviewers for their time to review our paper. Given the recent AC reassignment, we would like to briefly summarize the status of our paper to support the AC’s final decision.

We have provided comprehensive responses to all questions raised by four reviewers.

- Reviewer Nbom assigned a score of **8** and did not identify any major weaknesses.
- Reviewer pt64 gave a score of **8** and raised a core concern that the method may require multiple expert interactions, thus increasing data requirements and training costs. In our response, we clarified that our approach can demonstrate strong manipulation performance and semantic generalization using only 4k real-task trajectories, trained for 7,000 steps on 8×A100 GPUs. We believe we have effectively addressed this concern.
- Reviewer Cpwz considered the **soundness, presentation, and contribution of our work to be good**, with the main concern being whether using three experts for VLA is too heavy. In our response, we clarified that our method is not computationally heavy: unlike a 7B OpenVLA or the larger CogACT, our model has only 2.9B parameters, yet achieves significant performance gains and high execution frequency across multiple simulation and real-robot environments. Since Reviewer Cpwz did not respond to our rebuttal, we believe our clarification regarding the UniCoD model has addressed the concern.
- Reviewer 4’s core concern centers on novelty. We believe the reviewers have some misunderstanding regarding the paper’s innovations. In our response, we clarified that our work contributes three key innovations:

  1) We are the first to apply a mixture-of-transformers (MoT) to three heterogeneous tasks, while simultaneously leveraging world modeling to facilitate policy learning.

  2) We combine MoT with continuous-space world modeling, and we validate that learning world dynamics within a vision space rich in semantic information markedly improves generalization to unseen objects.

  3) We introduce a new training paradigm: unlike prior VLA pretraining strategies, our pretraining does not rely on low-level action signals. Instead, it uses the MoT architecture to learn discrete, language tasks alongside continuous-space world modeling—an approach that is crucial for semantic generalization in real-world scenarios.



Sincerely,

The Authors

---

### Note · Authors · 2026-01-31

I have read and agree with the venue's withdrawal policy on behalf of myself and my co-authors.

---

### Meta-Review · Area_Chair_nUM6 · 2026-01-10

**Summary:**

This paper proposes a VLA policy that couples discrete tokens with continuous predictions, and find that when learning semantic reasoning, planning, and explicit future representations jointly, the policy will have better performance.

The initial scores are quite divide. Two reviewers give positive scores, while the other two give negative. The concerns include:
- The contribution is straightforward and the novelty is limited.
- The baselines are limited. Some related works are not discussed.
- The writing could be further improved.

Given this situation, I have read the paper as well as the reviews. I believe the paper has two main drawbacks:
1. I agree with Reviewer TXuu and Cpwz that the novelty of the paper is limited. Adding multiple losses to train one model jointly is not a new idea, which has been adopted in many representative works such as JEPA (MSE), pi0 / pi0.5 (CE+Continuous loss).
2. In real-world experiments, the reported performance improvements of the proposed model over existing baselines do not demonstrate clear statistical significance, which makes it difficult to draw strong conclusions about its practical advantages.

**Reviewer Concerns:**

The concerns on novelty will still exist after the rebuttal, as no reviewer posted response.

**Reviewer Scores:**

All reviewers will keep their original score, as none of the reviewers engaged in the rebuttal.

---

### Decision · Program_Chairs · 2026-01-26

Reject